# Agricultural Robot-Centered Recognition of Early-Developmental Pest Stage Based on Deep Learning: A Case Study on Fall Armyworm (*Spodoptera frugiperda*)

**DOI:** 10.3390/s23063147

**Published:** 2023-03-15

**Authors:** Hammed Obasekore, Mohamed Fanni, Sabah Mohamed Ahmed, Victor Parque, Bo-Yeong Kang

**Affiliations:** 1Department of Robot and Smart System Engineering, Kyungpook National University, Daegu 41566, Republic of Korea; 2Production Engineering and Mechanical Design Department, Mansoura University, Mansoura 35516, Egypt; 3Department of Mechatronics and Robotics Engineering, Egypt-Japan University of Science and Technology, Alexandria 21934, Egypt; 4Electrical Engineering Department, Assuit University, Assuit 71515, Egypt; 5Department of Modern Mechanical Engineering, Waseda University, Tokyo 169-8050, Japan

**Keywords:** agricultural robotics, classification, deep-learning, detection, fall armyworm

## Abstract

Accurately detecting early developmental stages of insect pests (larvae) from off-the-shelf stereo camera sensor data using deep learning holds several benefits for farmers, from simple robot configuration to early neutralization of this less agile but more disastrous stage. Machine vision technology has advanced from bulk spraying to precise dosage to directly rubbing on the infected crops. However, these solutions primarily focus on adult pests and post-infestation stages. This study suggested using a front-pointing red-green-blue (RGB) stereo camera mounted on a robot to identify pest larvae using deep learning. The camera feeds data into our deep-learning algorithms experimented on eight ImageNet pre-trained models. The combination of the insect classifier and the detector replicates the peripheral and foveal line-of-sight vision on our custom pest larvae dataset, respectively. This enables a trade-off between the robot’s smooth operation and localization precision in the pest captured, as it first appeared in the farsighted section. Consequently, the nearsighted part utilizes our faster region-based convolutional neural network-based pest detector to localize precisely. Simulating the employed robot dynamics using CoppeliaSim and MATLAB/SIMULINK with the deep-learning toolbox demonstrated the excellent feasibility of the proposed system. Our deep-learning classifier and detector exhibited 99% and 0.84 accuracy and a mean average precision, respectively.

## 1. Introduction

Farmers suffer annual damage of over 35–40% from insects, plant pathogens, and weed pests [1,2]. Studies show that 94% of all known insect species are harboured outside water [3], and 70% of some insect families live in the soil before taking to the air as adults [4]. Thus, many pests and insects that infest plants already exist on farms, either as insects in a diapause state or as residue transmitted from the immediately previous planting. While monocropping is at a higher risk of infestation, multi-cropping systems are not entirely immune, especially to pests that infest various crops. For instance, the fall armyworm (FAW) does not require diapause; however, as FAW has over 40 crops in its diet, farmers must have diverse knowledge to prevent their growth in previous harvests. Preventing this problem can help save sufficient food to feed more than 3–4 billion people worldwide [1,2]. One of the methods to mitigate these potential infestations is using natural enemies, such as birds, which feed on insect pests effectively during land preparation for subsequent planting. Unfortunately, the chemical usage history of farms has endangered their lives and hampered the eco-friendly effectiveness of these natural enemies. Because of their bulk efficacy, methods based on chemical application are the most popular in practice. However, a direct consequence of their extensive and imprecise use is an imbalance in the ecosystem [5,6].

Over the years, smart off-the-shelf robotic sensor selection with improvements in machine vision techniques has been strongly associated with incremental progress towards reducing the use of chemicals in farming [7], from spraying through precise dosing to directly rubbing pesticides and herbicides onto the infected crops [8,9,10,11,12,13]. Functional agrobots [14,15,16] pioneering these advancements exhibit low versatility. Consequently, they cannot be utilized for chasing fast-moving (flying, crawling, or hopping) insects, spotting all types of insect pests in any background or lighting, and precisely neutralizing insect pests to perfection by spraying a chemical.

Due to the poor adaptability of agrobots and the less agile insect pest stage, this work suggests employing an off-the-shelf RGB stereo camera with deep learning to identify insect pest’s larva stage to avoid chasing after adult insects. This shift in the hunting timeline is also because the damage caused by some of these insect pests during their early developmental stages exceeds that caused during their adult stages. As a result, as a step toward robust all-pest recognition at the larva stage, a dataset containing images of FAW at the larva stage was gathered, and a deep-learning insect classifier and detector model were trained. These two deep-learning algorithms model human peripheral and foveal line-of-sight vision mechanisms as a trade-off between computational speed and accuracy because current deep-learning classifier models are typically less computationally intensive than deep-learning detector models. This means that after several streams of inferences on the classifier model have signalled the presence of an insect roughly, the insect detector model is only employed for the accurate localization of the appropriate robot’s insect pest neutralization mechanism. The RGB stereo camera sensor adopted is suggested to be mounted at the front of a robot to provide the data for the deep-learning models and insect pest localization estimation.

Therefore, this study presents a robot-centred early developmental stage of insect pests recognition by combining deep-learning object classification with detection for a proper neutralization of harmful agricultural insect pests and, consequently, with a reduced chemical footprint. Hence, the significant contributions of this study are as follows: First, an opensource dataset for the early developmental stage of insect pest, FAW, in this case. Second, accurate larvae stage insect pest recognition (classification and detection) models based on deep-learning, in contrast to hand-engineered classical object detection. Finally, the bioinspired vision system for robots eliminates the need for them to halt frequently before detection.

This paper is divided into five sections. Following the introduction in Section 1, a brief insight into related studies is presented in Section 2. The methodology of the research is introduced in Section 3, which briefly describes the data source, adopted sensor, deep-learning classification, detection architecture, and details of the proposed early developmental stage recognition mechanism for robot. Section 4 discusses the results obtained from the vision architecture, and cosimulation with the robot model in CoppeliaSim and MATLAB/SIMULINK with deep-learning architectures. Finally, Section 5 presents the conclusions and recommends possible directions for future studies.

## 2. Related Works

Since immemorial, pests, insects, and plant pathogens have been causing havoc on farms. Despite much research directed toward developing different traditional control methods, early-age farmers still get over 35% of all annual sustenance devastated by these agents [2], directly or indirectly. These methods include agricultural, physical, organic, chemical, and biological [17]. Modern pest management methods have become more pertinent as some traditional methods affect the ecosystem and, most importantly, the health of its inhabitants. This section presents trends adopted for sensor data processing toward minimizing chemicals in farming for robot-based agricultural insect pest control.

Prior primitive robotic insect pest control results utilize mobile robots for the bulk spraying of pesticides without prior detection [18]. This technique saves farmers from direct chemical contact; however, the robot’s capability was limited to only keeping farmers on the farms safe (either in greenhouses or open fields). The need to protect the entire planet from the effects of agricultural chemicals increases as the crop are destruction caused by insect pests is kept under control. Recent study findings intelligently link smart off-the-shelf sensor choices with efficient algorithms to handle the data. In [19], classical computer vision algorithms such as multi-template matching and the contour extraction of RGB camera images are employed on an agrobot to detect Pyralidae insects. In particular, a hyperspectral image sensor with partial least squares-discriminant analysis produced results for pest and disease detection in the laboratory and field as 66.4% and 59.8%, respectively. This sensor was created for an agricultural robot platform with a wide array of sensors [20]. Linear and quadratic discriminant analysis and support vector machines were also experimented on, but they are required to attain precision. Lucet et al. [21] proposed a mobile robot for pesticide-free aphid crop pest control with a ZED mini RGB-D sensor; this sensor provides stereo camera images for the YOLOv4 trained on a unique aphid dataset. Upon detection on the 2D RGB image (XY), the ZED mini SDK is used to localize the aphid pest approximately in 3D space (XYZ), allowing visual serving control to guide the proposed laser-based neutralization precisely. Meshram et al. [22] present a thorough analysis of sensing, target detection, and pesticide spraying for agrobots.

Detecting insects is quite challenging, considering their tiny sizes for detection algorithms and their agility at the adult stage for precise neutralization. Lucet et al. [21] also experimented with stereo sensor mount distances from ground and output image size sub-sampling to arrive at <30 cm and 800 × 800 pixels in 4 or 16 sub-parts for 0.73 sensitivity, respectively. Consequently, some researchers have employed an indirect method for controlling insects by limiting the growth of weeds on farms. Weeds are known to breed both beneficial and nonbeneficial [23] insects. To minimize chemical usage, ref. [8] applied image and spectral analyses to differentiate weeds from crop plants to precisely dose herbicides on different weeds. Similarly, ref. [9] directly applied further reduced chemical usage to the vascular tissue of weeds by using the robot’s end-effector to cut the weed stems and wipe the chemicals on its cut surface. Moreover, ref. [10] developed a classical computer vision algorithm for spraying the pest-affected regions of plants in a greenhouse.

Obviously, the effectiveness and efficacy of the recently proposed approaches were strongly coupled with the advancements in vision algorithms. Recent advances in deep-learning-based object detection have proposed frameworks in two broad categories: one-stage detectors, such as you only look once (YOLO) [24] and its derivatives [25,26,27], and two-stage detectors, such as region-based convolutional neural networks (R-CNNs) [28], and their variants [29,30,31]. These categories have introduced new techniques and strategies to achieve good tradeoffs between speed and accuracy.

Alvaro et al. [32] presents a robust deep-learning-based detector for the real-time recognition of tomato plant diseases and insect pests on the farm using three main detectors: faster R-CNN, the region-based fully convolutional network (R-FCN), the and single-shot multibox detector (SSD) with varieties of base networks as feature extractors. These proposed techniques can effectively recognize nine diseases and insect pests under complex farm scenarios. Ard et al. [33] also used the faster R-CNN for detecting and counting insects on yellow sticky traps in tomato crops. Similarly, ref. [34] applied the Yolo-V3 model using image spatial pyramid pooling as the multiscale feature detection on a custom-built tomato disease and insect pest dataset. Their detection results show accurate and quick localization and categorization of diseases and insect pests of tomatoes in the real natural environment. The recognition of rice plant diseases and insect pests has also been performed using a deep CNN and evaluated on both still images [35] and video streams [36] with decent accuracy and speed. Deep CNN is a data-oriented technique in which many researchers augment their local source insect dataset by sourcing online images for robustness and better model generalization. Wang et al. [37] investigated ten insect pests mainly affecting tea plants with images sourced online, one of which was the *Spodoptera exigua* larva. Likewise, ref. [38] achieved 87% accuracy on the Caffe framework with an augmented dataset of paddy (rice) pests and diseases sourced online. The DockWeeder project aimed at removing a troublesome weed, called broad-leaved dock (*Rumex obtusifolius* L.), and deployed a DockWeeder prototype robotic platform as a data collection platform for training deep learning models [39,40]. Additionally, the deep-learning model results performed better than their initial classical computer vision approach [41].

One common thing among the approaches mentioned above is that they either work directly on the insects while aiming at reducing chemical usage or adopt an indirect approach by working on weeds. As a result, we suggest employing a deep-learning-based method to strategically hunt the insect pests’ larvae stage to serve as a means for simple robot configuration. Similar to [21], we also suggested using an off-the-shelf stereo sensor for deep-learning insect pest recognition and localization. The strategy takes the merit of the robustness of two deep-learning algorithms, a deep-learning classifier and a deep-learning detection model, to classify and detect insect pests at the early stage (i.e., larvae) during preplanting, as well as to precisely provide localization for the agrobot to localize above the insect pest to neutralize with any means that minimizes the quantity of chemicals to be used. For instance, with precisions such as this, refs. [42,43] proposed using physical or mechanical control methods, such as handpicking; vacuuming; modifying environmental conditions, particularly heat and humidity in the case of greenhouses, solarization or steam sterilization; and visual or physical deterrents for open fields.

## 3. Proposed Methodology

### 3.1. Site Location

This study sourced the insect pests from West Africa, specifically Southwest Nigeria (8.1574° N, 3.6147° E). The chosen insect pest for the case study was the fall armyworm (*Spodoptera frugiperda*). Fall armyworm insects pose a significant threat to agriculture and are independent of any crop. This insect affects over 40 plant species [44] (Figure 1a shows their lifecycle ), ravaging Central and West Africa since early 2016. It almost invaded sub-Saharan Africa within two years and was confirmed in the Asian region in July 2018. By October 2019, the Asian region had included China, the Republic of Korea, and Japan. Recently, Australia and the United Arab Emirates were included between January and May 2020. Figure 1b shows a detailed overview of the geographic distribution.

### 3.2. Data Collection and Image Sensors

In sourcing for this insect pest, strict adherence to the description established in [45] was ensured, with close verification by professional entomologists and agriculturists. Owing to seasonal limitations, we only collected 862 images and some videos of this insect in different lighting, orientations, and backgrounds for the insect dataset. Many were taken from farms, markets, and infested products using mobile phone cameras. These image data are further preprocessed for the training of each deep learning technique.

**Figure 1 sensors-23-03147-f001:**
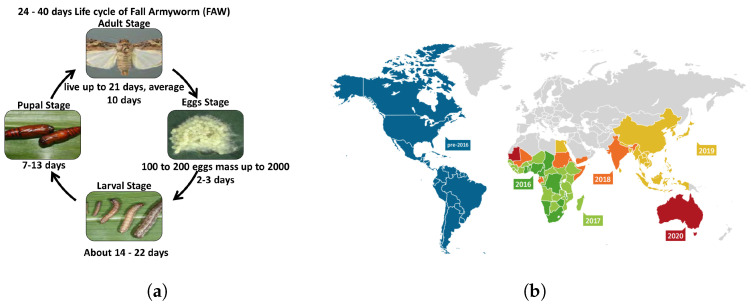
Lifecycle and geographic distribution of proposed insect pest case study (fall armyworm). (**a**) Lifecycle of fall armyworm [46]; (**b**) Geographic distribution as of May 2020 [47].

As previously noted, we proposed an off-the-shelf stereo camera as the data source for the deep learning models of the insect pest larvae scouting robot. A stereo camera is one of the sensors that provides means to localize objects in images in a 3D world frame. It is an abridged model of human binocular vision using two or more cameras separated at a certain baseline distance to create binocular disparity necessary to estimate the depth of objects in the captured image. For instance, given that some insect pests are present in the left and right cameras of a stereo camera, a deep learning detector can produce corresponding n bounding boxes for each detected insect pest $\left(x_{n}^{L},\;y_{n}^{L},\;h_{n}^{L},\;w_{n}^{L}\right)$ and $\left(x_{n}^{R},\;y_{n}^{R},\;h_{n}^{R},\;w_{n}^{R}\right)$, respectively. The potential 3D location of each insect pest in the world frame is $X_{n},\;Y_{n},$ and $Z_{n}$. Accurate camera calibration and vision correspondence are important to develop reliable estimation. Many off-the-shelf stereo sensors, such as ZED series (StereoLABS Inc., San Francisco, CA, USA), RealSense^™^ Depth camera series (Intel^®^, Santa Clara, CA, USA), and Bumblebee 2 FireWire (Flir, Santa Barbara, CA, USA) provide SDK/API for out-of-the-box 3D world localization. Of course, custom pipeline implementation is also possible by choosing from scholarly techniques [48,49]. For example, in this work, scale-invariant feature transform (SIFT) and speed-up robust feature (SURF) algorithms [50,51] are experimented with for the correspondence of vision, while the camera calibration adopted the use of ChArUco marker approach [52].

A front-pointing RGB stereo camera with calibration is the suggested sensor for the prospective agrobot; it produces a stream of frames of images from each angle. This stream from the stereo camera was divided into two unequal sections: farsighted and nearsighted (Figure 2). The farsighted section, fed into a deep-learning classifier, is referred to as the peripheral vision; further, the nearsighted section, fed into the deep-learning detector, is referred to as the foveal line of sight. This is a method adopted to allow a robot to classify whether an object of interest (i.e., a harmful insect) exists in the view from afar while still in motion by leveraging on the fact that objects captured by a front-pointing camera in the direction of motion (of a forward moving robot) will first appear in the farsighted section. Therefore, streaming the less computationally expensive deep learning classifier on the farsighted section to determine when to initiate the execution of the deep learning detector on the nearsighted section would be a fair trade-off between the robot’s smooth operation and localization precision.

### 3.3. Proposed Robot-Centered Deep-Learning-Based Insect Pest Scouting Method

The proposed robot-centred insect pest recognition is modelled to scout for insects on the farm by following farm implements to look out for exposed insect pests by taking advantage of the robustness of two trained deep-learning algorithms (deep-learning classifier and detection models) while actively running the classifier and, subsequently, the detector. This relative insect pest scouting method of the robot is inspired by the flocking of *Bubulcus ibis* (cattle egret) around cattle and farm machinery to feed on insects during preplanting operations (e.g., ploughing and harrowing). Although chemical activities on farms divert them away, this feeding technique has been shown to be 3.6 times more effective [53] because many insects diapause underneath the soil to await favourable periods. Farm equipment used in preplanting operations helps uncover insect larvae buried beneath the earth. Thus, it allows farmers to get rid of potentially harmful insects at their early developmental stages before actual plant infestation. Thus, the proposed system is recommended to follow farm machinery during preplantation to optimally detect insect pests with the data from the stereo camera mounted in front of a robot.

Similar to the bioinspiration behind the proposed scouting technique of the robot, the vision mechanism adopted for the proposed insect pest recognition was also bioinspired by how humans use their eyes when looking for something specific. Humans do not stop looking at everything; they only stop for explicit confirmation after some senses have voted positively for a specific object. The human eye’s peripheral vision is blurry but hints at what is in view. Foveal vision is a clear and precise view of the eyes (Figure 2). If any specific object of interest is sighted in the peripheral view, the foveal view turns toward it. Here, peripheral vision serves as the object’s classifier in view of actual recognition and detection by foveal vision [54]. On a regular embedded device CPU, the deep-learning detector can only achieve <2 frames per second (fps), whereas the deep-learning classifier can achieve up to 8 fps. Because the classifier can execute faster fps by drawing inspiration from the human vision mechanism, the proposed technique is to scout harmful insects by running a deep-learning insect classification while traversing the farm at a steady speed (usually robot). Thus, any positive sensing from the classifier determines what catches the robot’s attention for proper and precise detection with the deep-learning harmful insect detector model. Our specific interest lies in the insects’ larvae stage.

This action leverages the fact that current classifiers are generally faster than current detectors. If the object of interest is found, the robot stops to feed the nearsighted section into the deep-learning detector to obtain bounding boxes, and the corresponding three-dimensional (3D) position is estimated. A SIFT algorithm [50,51] was applied to ensure the correspondence of each insect’s 3D position in both frames from the stereo camera. The detector obtains the output positions with object correspondence, and the necessary transformation factor can be applied to localize optimally above the insect pest to utilize the appropriate neutralization method. After the neutralization of detected insects, the robot platform continues to scour the farm for more insect pests. A detailed overview of this process is shown in Figure 3 and Figure 4, which show the flowchart of the whole workflow. The stereo cameras are the left and right cameras, and $\left(x_{n}^{L},\;y_{n}^{L},\;h_{n}^{L},\;w_{n}^{L}\right)$ and $\left(x_{n}^{R},\;y_{n}^{R},\;h_{n}^{R},\;w_{n}^{R}\right)$ are their corresponding n bounding boxes detected by the insect detector, respectively. $X_{n},\;Y_{n},$ and $Z_{n}$ are the locations of n number of insects detected with respect to the robot.

### 3.4. Deep-Learning Architecture for Insect Pest Classification and Detection

Deep-learning classification is the trending state-of-the-art machine-vision technique that gives the tendency of an image belonging to a particular class or category using CNN. Given an input image, the artificial neural network associates the image with a class label with a certain probability [55] (Figure 5). Deep CNNs typically comprise a sequence of CNN layers, pooling layers, nonlinear activation layers, and fully connected layers (FC layers) (Figure 5). With a nonlinear activation function connecting each layer, the convolutional layer convolves the learned kernels over the input image to generate a feature map. During training, these kernels are jointly optimized using different optimizers, ranging from simple stochastic gradient descent (SGD) [56] and RMSProp to ADAM [57]. These complexly convolved layers, with the final FC layer having a softmax layer, define the deep CNN classification architecture. For example, AlexNet [55] contains five convolutional layers, three max-pooling, and three FC layers. Each convolutional layer is followed by a rectified linear unit (ReLU) [58], a nonlinear activation function.

Herein, the approach adopted was transfer learning with a preloaded ImageNet weight. Transfer learning deals with repurposing existing successful deep-learning models by fine-tuning the model on a small dataset of the new class(es). Table 1 lists the experimental deep-learning classifier architectures with their respective pretrained weights initialized for fine-tuning. While these architectures commonly use CNN layers with varying depths, many implement unique interconnections, dimensionality reductions, and residual connections. Note that GoogleNet proposed a 22-layer deep network with nine inception modules. The architecture features a small network (inception module) within a larger network [59]. In addition, one-by-one convolutional layers were used for dimensionality reduction and feature aggregation. Inception V3 [60] is a CNN architecture that is 48 layers deep by heavily using the deep interconnections of inception modules.

The ResNet architecture [61,62] introduces residual layers and skip connections to solve the problem of a vanishing gradient, which may stop the weights in the network from further updating or changing. ResNet50 and ResNet101 indicate 50 and 101 deep layers, respectively. The SqueezeNet architecture utilizes the concept of a fire module, which contains three filters of one-by-one-sized feed into an expanded layer (four filters of one-by-one and three-by-three sizes) [63]. However, AlexNet [55] and visual geometry groups (VGGs) [64] are not unique in their architecture. The only significant difference lies in their depth. Figure 5 shows the two possible instances of our insect classifier taking in only the farsighted section, both in the presence and absence of the insect. The in-between is the most accurate base network used from the other tested pretrained architectures in Table 1. To support our two classes of outputs, the current output layer is changed.

**Table 1 sensors-23-03147-t001:** List of the ImageNet pretrained models optimized for feature extraction and insect classifiers.

Method	Description
AlexNet [55]	This architecture has a convolutional neural network that is eight layers deep. It takes an input image size of 227 × 227 × 3.
GoogleNet [59]	This model architecture is tiny compared to AlexNet and VGGNet. It uses micro-architectures such as inception modules. The expected input image size is 224 × 224 × 3.
Inception V3 [60]	This model is a 48-layers-deep convolutional neural network. It has an input image size of 299 × 299 × 3.
ResNet50 [61]	This architecture leverage on a residual module to train convolutional neural networks upto depths previously impossible. It has 50 layers of convolutional neural network layers. It was trained with an input image size of 224 × 224 × 3.
ResNet101 [62]	This is a 101-layers-deep convolutional neural network deep variant of ResNet50. It takes input image size of 224 × 224 × 3.
SqueezeNet [63]	This network has an image input size of 227 × 227 × 3; it is 18 convolutional layers deep.
VGG16 [64]	This architecture is a 16-layers-deep convolutional neural network. It expects an input image size of 224 × 224 × 3.
VGG19 [64]	This is a 19-layers-deep variant of VGG16. It expect an input image size of 224 × 224 × 3.

Deep-learning object detection classifies the content of an image as a classifier and draws a bounding box over the object’s location in the image (Figure 6). As previously mentioned, deep-learning object detectors exist in two major paradigms: one-stage detectors, such as YOLO [24,26,27] and SSD [25], and two-stage detectors, such as R-CNN [28,29,30], and its variants [31].

Herein, a two-stage detector is employed, specifically the faster R-CNN algorithm, because it is generally more accurate than its one-stage counterpart (Figure 6). The faster R-CNN components include the base network (pretrained feature extraction), anchors, region proposal network (RPN), region of interest (ROI) pooling, and R-CNN. The base network is a typical pretrained CNN classifier; specifically, the classifier is pretrained on the ImageNet dataset. This typically serves the purpose of feature extraction (Figure 6) and gets fine-tuned by transfer learning for the specific objects to detect during the end-to-end training. The base network is often modified to be fully convolutional, that is, with no FC layers. Many deep learning-based object detection architectures have been proposed with a particular feature extractor, such as VGG and residual networks (ResNet). However, in this study, we focus on tuning the feature extractors of the faster-RCNN by experimenting with different pretrained classifier architectures, in contrast to those initially proposed with the architecture as in [32]. This approach was implemented to investigate the possibility of obtaining maximum performance.

Table 1 presents the feature extractors selected for the experiments. Concerning the anchors, with regard to the original faster R-CNN publication, the input image was discretized into a stride of 16 pixels surrounded at the centre by 64∗64, 128∗128, and 256∗256 bounding boxes, each with a scaled aspect ratio of 1–1, 1–2, and 2–1 generated to yield a total of nine anchors. The RPN is then used to determine the location of a potential object, that is, if the ROI is either a background or foreground. The dimension of one of its outputs is 2∗K, where *K* is the total number of anchors, and the other two hold the probabilities of the foreground and background. The second output has a dimension of 4∗K ($\delta_{x},\;\delta_{y},\;\delta_{width},\;\delta_{height}$), which contains a bounding box regressor that can be used to adjust the anchors to fit the object better than its surroundings. Before ROI pooling, the K anchors are first passed through non-maxima suppression to eliminate multiple overlapping bounding boxes, resulting in N proposal locations. Thus, the main goal of ROI is to accept all the proposals from the RPN module and crop the output feature vectors from the convolutional feature map. Finally, the R-CNN is simply a FC layer that predicts the final class label through the softmax classifier and refines the bounding box for better accuracy through the bounding box offset regressor.

As mentioned earlier, the chosen insect pest was the fall armyworm (*Spodoptera frugiperda*). For classification, we preprocessed the dataset by closely cropping the images and annotating them as a binary class (FAW or no-FAW). The dataset for the negative class was a sample of wormlike insects ( not FAW) from the iNaturalist dataset [65]. Figure 7 shows samples of images in the FAW dataset. The raw images were annotated by drawing ground-truth bounding boxes around the insect location for the deep-learning detector. The videos were reserved for further testing.

In summary, the insect faster-RCNN detector was fine-tuned on an insect dataset. With the feature extractor being part of the tuneable parameters, different base networks are tried, unlike the network initially proposed with faster-RCNN, to select the best performance in terms of their mean average precision (mAP) [66]. Similarly, to choose our insect classification architecture, we fine-tuned the different classifiers in Table 1 to optimize the detection speed and accuracy of the insect dataset jointly.

## 4. Experimental Results

The dataset was divided into 70% training and 30% validation to train the faster-RCNN insect detector, whereas testing was performed using the video dataset. The training has four stages: training the RPN, training the fast-RCNN using the RPN, retraining the RPN using weight sharing with the fast-RCNN, and retraining the fast-RCNN using the updated RPN. The same epochs and learning rates were employed for each stage. All of the implementations were conducted by using MATLAB 2018b for training and robot control on an Intel(R) Xeon(R) Gold 2.30 GHz processor coupled with an NVIDIA Quadro P4000 GPU. Moreover, all experiments on the embedded device were performed on a 4GB RAM NVIDIA Jetson Nano.

### 4.1. Insect Classifier and Detector Performance

First, in choosing the classification architecture, we considered the classification categories in the ImageNet [67] dataset and observed that two of the 1000 classes were similar to those from our dataset. Our chosen insect biologically belongs to the phylum Arthropoda. Its larval stage has substantial visual similarity to the Nematodes and Platyhelminthes. Based on this information, we fed our datasets into classification architectures pretrained on ImageNet that are available in the MATLAB deep-learning toolbox, and they were largely activated only by the Nematode class. Therefore, our model strengthened the Nematode class by fine-tuning the class in our dataset with extensive data augmentation through cropping, random rotation, and random scaling. After fine-tuning for six epochs and a $3\times 10^{-4}$ learning rate, Table 2 presents the accuracy and inference speed for all architectures, both on the GPU of the personal computer (PC) and the embedded device. Testing accuracies of 99% and 87 and 24 fps on the GPU for PC and embedded devices, respectively, were achieved, with an emphasis on the most accurate and fastest classifier, VGG19. Hence, VGG19 was adopted for the farsighted sections of the robot.

Second, for the faster-RCNN insect detector, Figure 8 presents the results of the mAP of the detector after 250 epochs on the two classes from our dataset (SPODOPERA and background) for each base network. After dropping the poor-performing feature extractors, Figure 9 presents the same measure after 350 epochs. This approach hoped to achieve better performance; however, the insect detector based on GoogleNet suffered stagnation, whereas ResNet101, VGG16, and VGG19 were reduced. In each case, however, VGG16 outperformed all of the other base networks experimented with, perhaps because it was initially its native proposed base network. Thus, we adopted the faster-RCNN with VGG16 as a base network trained at 250 epochs as the insect detector for the nearsighted robot section. Figure 10a–c show some successfully detected instances, whereas Figure 10d shows a wrongly-detected instance.

Finally, we experimented with SURF and SIFT image-matching algorithms to ensure correspondence of the location in each camera view based on the bounding box output of the detector. Table 3 shows the number of features and matches obtained by applying the two algorithms to the stereo camera’s left (L) and right (R) views. Figure 11 presents the visualization of the instances in Table 3. The results show that SURF has a fast computation time but poor output matches. In contrast, SIFT delivered better matches at the expense of computation time.

### 4.2. Cosimulation Results

The proposed technique is formulated to be used on agricultural robots. Therefore, the robot kinematics, dynamic model, and controller proposed in [68] are adopted to validate its usability. The robot is a mobile manipulator having a four-DOF RRRR manipulator with a skid-steering mobile base. For cosimulation with our proposed method, the robot model is ported to CoppeliaSim. CoppeliaSim (formerly V-REP) is a flexible simulation software that is useful for robot modelling and simulations, fast algorithm development, factory automation simulations, fast prototyping, and verification and as a digital twin [69]. It is available for free for academic usage and includes features modules and add-ons to integrate with many platforms, such as the robot operating system. Figure 12 depicts the imported model representing the visual elements on the agricultural terrain in the CoppeliaSim scene. Two front-mounted perspective vision sensor models provided in CoppeliaSim were also employed to model the chosen ELP 1.3 megapixels ELP-960P2CAM-LC1100 OV9715 [70] off-the-shelf stereo sensors. When the classifier foresees the appearance of potential insects in the farsighted phase of the simulation, the mobile platform only stops for exact detection. Figure 13 reveals the insects detected by the insect detector during each stoppage in the cosimulation. Each shot contains the stereo camera’s stitched left and right views. The cosimulation agrobot travels a maximum distance of 10 m in 65 s. Its coverage is dependent on the population density of the insect pests spawned in the simulation, in addition to the computing cost. Nevertheless, during the simulation, the robot stopped four times. One insect was detected at the first stoppage, whereas two were detected in the second instance. Furthermore, one insect was detected in the third and fourth instances. During these stops, the method also provides the information necessary to localize above each detected insect to apply an appropriate neutralization method, after which the subsequent moving and classifying periods can be triggered.

Although the SIFT algorithm result is not so apparent from the shots in Figure 13, by observing the stream in the Appendix A, it is clear that not all detections trigger a corresponding match on both stereo views. Therefore, the SIFT algorithm further assists the insect detector in filtering out, thus creating more stable localization information.

### 4.3. Discussion

In this paper, we have described a robot-centred deep learning-based insect pests larvae recognition with an RGB stereo camera as the sensor. A farsighted event acts as a trigger to feed the nearsighted section of the cameras’ two simultaneous pictures, which are divided into farsighted and nearsighted sections. Because the farsighted section is fed into our deep learning insect classifier, the lower boundary for the vertical limit is constrained by the minimum acceptable image input size of the chosen pretrained model in Table 1, VGG19 (224 × 224 × 3). However, the upper bound limit could be constrained through trial and error, which synchronizes robot deceleration with the maximum robot speed in image space. We frequently avoid abrupt robot stopping; therefore, this synchronization is crucial for stability. As a result, during uniform deceleration, the presence of insect pests that triggered the positive class of the insect classifier (in the farsighted section) would have translated to the nearsighted section.

We opensource our FAW dataset for a possible extension to other harmful agricultural pests larvae. The usual labeling for classification and detection task convention is followed throughout the entire dataset annotation. We closely cropped out a 1:1 image of only the FAW in a single image for classification, and bounding boxes were drawn over each FAW for detection purposes. All of the raw images contain larvae; however, some of these raw images contain multiple larvae. As a result, multiple bounding boxes exist in the detection annotation for some images. Hence, both classification and detection data have 862 larvae each (862 images and 862 labelled bounding boxes, respectively). The dataset also contains 109 s duration videos reserved for testing.

This work’s main objective is to introduce an additional layer of precision to existing deep-learning agricultural pest recognition. According to the trendline, researchers have used deep CNN technologies to further improve robot precision, which has the effect of lessening the chemical impact via precise localization and dosage. Based on one of these deep CNN techniques, the Australian Center for Field Robotics created a robot for intelligent perception and precision applications (RIPPA) to detect weeds from an image sensor and remove them mechanically [16]. A similar strategy was employed by BoniRob, whose “ramming death rod” aids in mechanically eliminating weeds and plants when an AI algorithm detects them [14,15]. A deep learning-based intelligent spraying system with semantic segmentation of fruit trees in a pear orchard created a SegNet model with 83.79% accuracy. It further refines the segmentation by incorporating the depth information to separate the background. Consequently, actual farms are using fewer pesticides [71].

This work aims to show the feasibility of the proposed approach; however, we also comprehend that there is a need for significant improvement to translate competitively with the existing spraying or dosing technique on real robot hardware. For example, our result in Figure 13 experimented on a low insect pest population density, where the insect pests are widely spaced on the field. Nonetheless, we propose two worst-case scenarios that would require major exceptions in the field. One case is when insects are in the farsighted section but not in the nearsighted section, given that the robot has already stopped. In this situation, our suggested course of action is reduced to intermittent start/stop operations until the insects reach the nearsighted area. The other case is when there are insects in the near section but not in the far section, given that the robot is still in motion. In this situation, such insects are likely to be missed. Moreover, the capacity to recognize neutralized larvae is crucial for preventing recurrent detections. In our simulation setup, we handled this by assuming that the insect pests are always stationary. Therefore, every time a halt is made, bug pests are locally identified.

Nevertheless, the proposed technique still has significant usefulness, either whole or piecemeal. Overall, it can reduce or eliminate the need for pesticides, allowing farmers to make greater use of the pest larvae that have been eliminated. However, when decomposed into piecemeal, one strategy is to use the deep learning models (with or without robot) that focus only on the early developmental stage because they hold significant threats than the adult stage. Another strategy is to speed up the functioning of agrobots to use the two models as a tradeoff. Others include the following of farm machinery during preplanting procedures to assist agrobots in exposing hibernating larvae and early neutralization before infestation.

## 5. Conclusions

This study proposed a deep learning-based extendable and robot-centred strategy for recognizing harmful agricultural insect pests. This approach uses RGB data from a stereo camera sensor, split into the peripheral and foveal line-of-sight vision to catch the insect pest young. This is achieved by focusing on their early developmental stage (larvae) during preplanting operations, inspired by a natural enemy called cattle egret. A simple mobile manipulator robot was adopted to validate the proposed simulation technique’s usability by mounting the sensor at its front. The peripheral vision and the foveal line of sight adopted the VGG19 classifier with 99% accuracy and the faster-RCNN detector with VGG16 as the base network, respectively. This base network outperformed other investigated base networks, with a mean average precision of 0.84. The SIFT algorithm delivered a better performance at the expense of computation time for matching stereo camera views. Notably, our cosimulation approach in CoppeliaSim and MATLAB (with relevant toolboxes) confirms the system’s feasibility, which is a step toward actual implementation. However, major advancements are needed to transfer competitively with the current spraying or dosing technique on actual robot hardware. Operating on a swarm of robots is a potential remedy, but it is very expensive.

An unavoidable design limitation of the suggested method is the relatively wide blind spot, that is, the inability to capture all ground views of the robot’s camera at once, which may limit the work volume of robots. This is the usual design limitation for obtaining a realistic robot manipulator design. However, this originally uncaptured vision can be compensated by having several overlapping robot passes in the field.

Nevertheless, the potential application of the proposed method for farmers is its capacity to minimize or prevent pesticide usage. Avoiding pesticides could translate to better use of the neutralized pest larvae, such as animal feed. However, a more futuristic approach is to implement an unmanned area vehicle [72] as challenged in the VyavaSahaaya, 2019 [73]. 

## Figures and Tables

**Figure 2 sensors-23-03147-f002:**
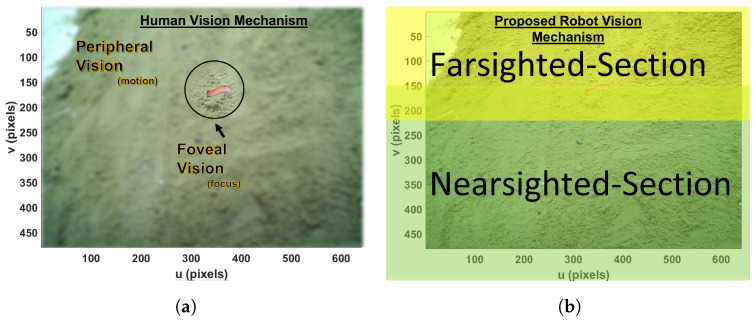
Modelling of human vision mechanism as the proposed insect pest larvae recognition vision mechanism. (**a**) Human vision mechanism. (**b**) Proposed robot vision mechanism.

**Figure 3 sensors-23-03147-f003:**
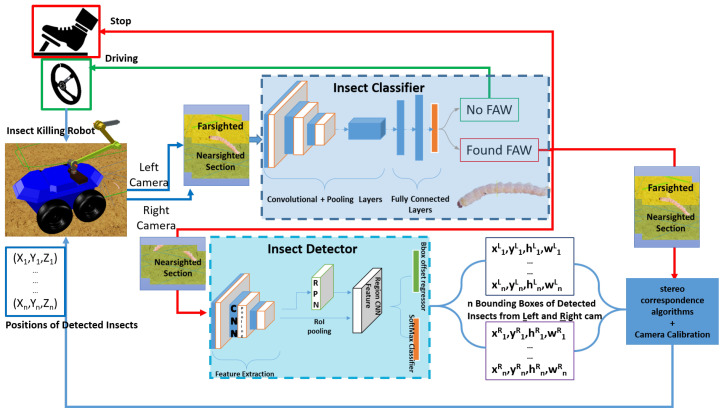
Proposed deep learning-based robot-centered insect pest larvae recognition overview.

**Figure 4 sensors-23-03147-f004:**
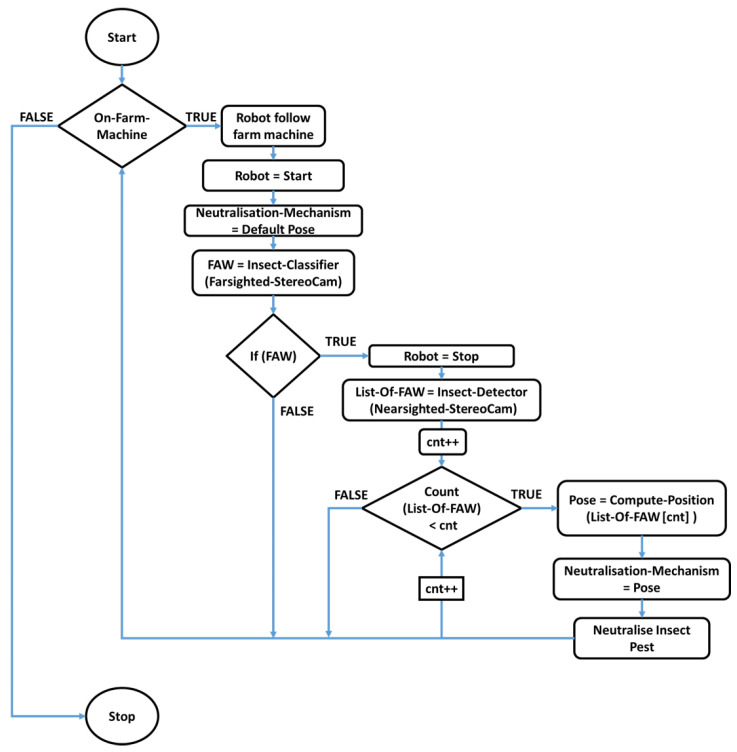
Proposed deep learning-based robot-centered insect pest larvae recognition flowchart.

**Figure 5 sensors-23-03147-f005:**
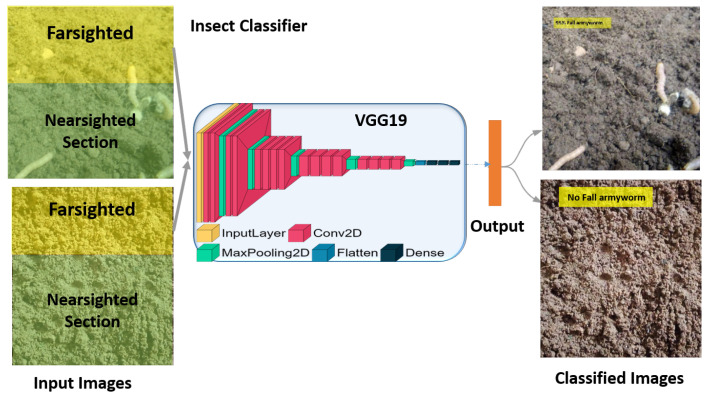
Network architecture of deep-learning insect pest classifier, with emphasis on VGG19 as the most accurate base network adopted for the farsighted section of the proposed recognition mechanism.

**Figure 6 sensors-23-03147-f006:**
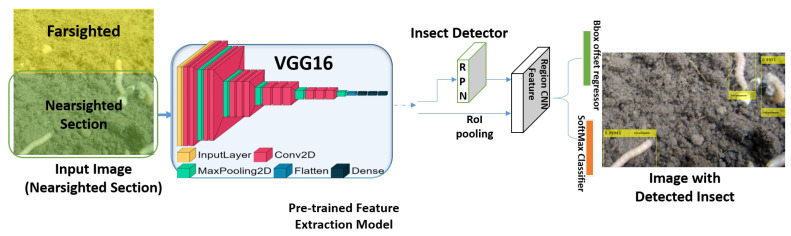
Network architecture of deep-learning insect pest detector with the best experimented pretrained model.

**Figure 7 sensors-23-03147-f007:**
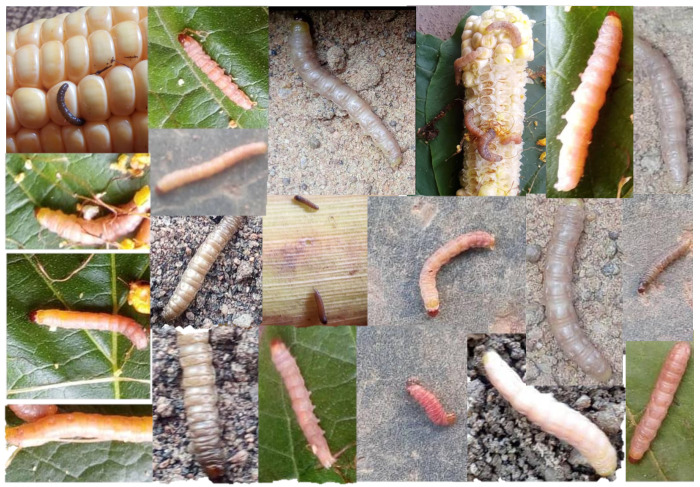
Samples of insects from our dataset. https://github.com/obasekore/Spodopera_DL_dataset, accessed on 5 March 2023.

**Figure 8 sensors-23-03147-f008:**
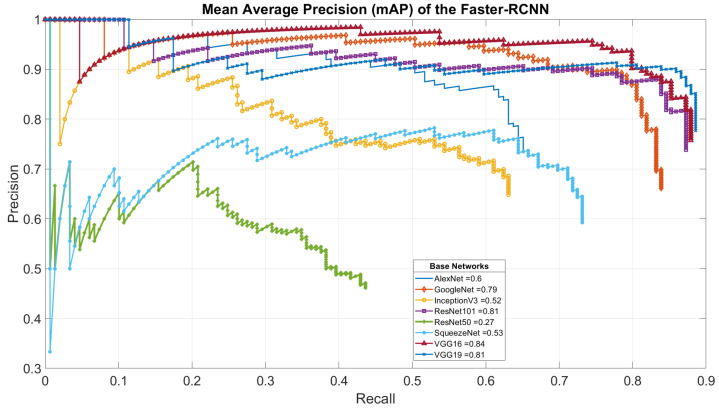
Mean average precision of our insect detector after training for 250 epochs with each feature extractor in Table 1.

**Figure 9 sensors-23-03147-f009:**
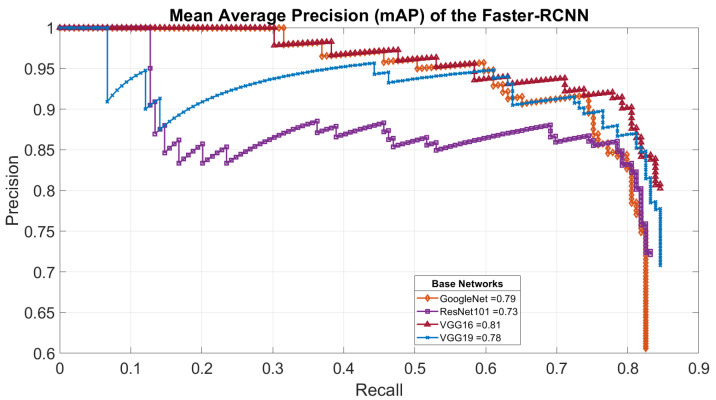
Mean average precision of our insect detector after training for 350 epochs with the most successful feature extractor.

**Figure 10 sensors-23-03147-f010:**
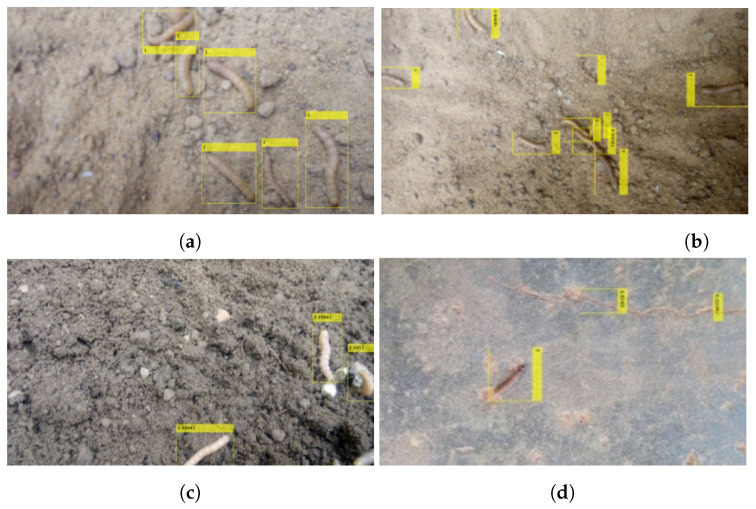
Images of successfully (**a**–**c**) and wrongly (**d**) detected insects using our proposed faster-RCNN insect detector.

**Figure 11 sensors-23-03147-f011:**
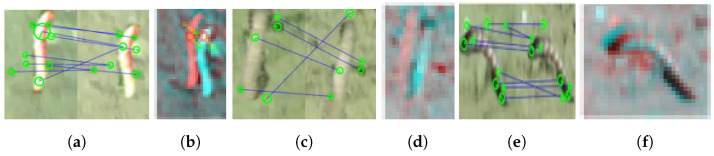
Outcomes of corresponding visual matches of SIFT (**a**,**c**,**e**) and SURF (**b**,**d**,**f**) are shown in Table 3. Each consists of two stereo camera views of the detected pest with matched features overlayed.

**Figure 12 sensors-23-03147-f012:**
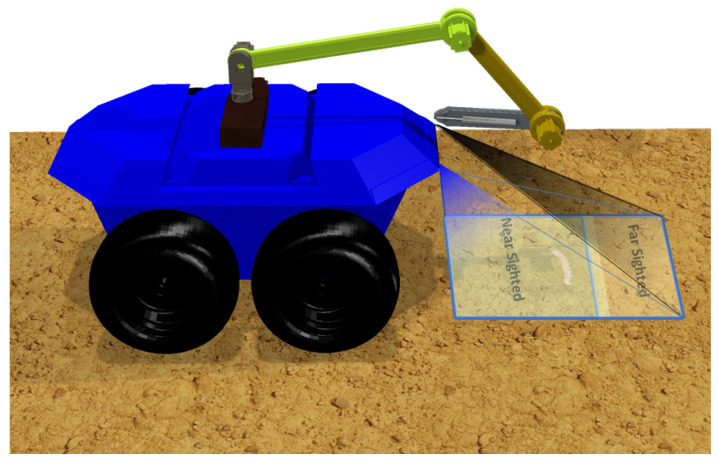
The adopted mobile manipulator in the CoppeliaSim scene for validating the proposed insect pest recognition mechanism.

**Figure 13 sensors-23-03147-f013:**
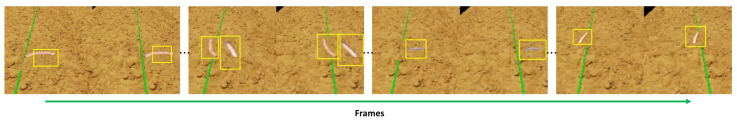
Results of the Faster-RCNN insect detector showing the detected insect pest during the cosimulation of the proposed recognition mechanism with robot in CoppeliaSim and MATLAB/SIMULINK.

**Table 2 sensors-23-03147-t002:** Validation accuracy after fine-tuning the proposed insect classifier.

Models	Accuracy (%)	Time on PC (fps)	Time on Embedded Device (fps)
AlexNet (%)	95.95	16	13
GoogleNet	94.99	14	14
SqueezeNet	96.92	34	23
Inception V3	93.26	2	7
VGG16	94.03	82	19
VGG19	99.04	87	24
ResNet101	93.83	3	6
ResNet50	94.03	4	10

**Table 3 sensors-23-03147-t003:** Comparison between SIFT and SURF to match the left (L) and right (R) of the stereo camera with the insect detector output.

SIFT				SURF			
Figure	L	R	Match	Figure	L	R	Match
Figure 11a	34	23	9	Figure 11b	9	7	3
Figure 11c	25	25	6	Figure 11d	7	3	0
Figure 11e	62	41	10	Figure 11f	12	10	0

## Data Availability

The fall armyworms data presented in this study are openly available in [Spodopera_DL_dataset] https://github.com/obasekore/Spodopera_DL_dataset github repository accessed on 5 March 2023. The negative class data utilized the wormlike samples from the publicly available iNaturalist datasets [65].

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
