# Peer review of "Agricultural Robot-Centered Recognition of Early-Developmental Pest Stage Based on Deep Learning: A Case Study on Fall Armyworm (Spodoptera frugiperda)"

_sensors, 2023, doi:10.3390/s23063147_

Round 1

Reviewer 1 Report

This paper describes a method to detect and locate worms in stereo RGB images, for its use in robots on the field. In general, the manuscript is clear and correctly organized, the research methodology is adequate and scientifically sound, and the results can be interesting for other researchers. However, the presented research seems to be in its initial stages; for example, the 3D location of the worms is not completed, and the robot presented is only a simulation.

I also have some more specific comments:

1. The title is not grammatically correct. You should say: "Based on Deep Learning".

2. In the affiliation of the authors, the country is missing.

3. The abstract looks more like an introduction than a summary of the work. You should rewrite the abstract, reducing the initial part and describing more the techniques on which the proposed system is based. For example, you only say that you use RGB cameras and "deep-learning algorithms". You should describe the type of networks, the options that have been considered, the dataset that have been used, etc.

4. Please, do not divide sentences incorrectly. For example, in lines 2-4 you say: "This ranges from the ability to use simple robot configuration to early neutralisation. Most importantly leading to reduced chemical pesticide input". The second part is not a sentence, but a part of the first sentence. So, you should not divide it with a point. You should say: "This ranges from the ability to use simple robot configuration to early neutralisation, most importantly leading to reduced chemical pesticide input". This is a very frequent mistake that you make throughout the entire article.

4.1. The same thing happens in lines 37-38: "Starting from spraying through precise dosing to directly rubbing pesticides and herbicides onto the infected crops". This is not a sentence, but a part of the previous sentence.

4.2. The same error in line 44, with sentence "Thus matching the low versatility...". This is not a sentence, but a part of the previous sentence.

4.3. The same in line 53: "Meaning after several...". You can say "This means that after several..."

4.4. The same with the sentence in line 109: "Weed-breed both beneficial and nonbeneficial [23] insects". That is not a sentence! Where is the verb? A sentence must have: subject + verb + predicate.

4.5. The same in lines 204-205: "While the camera...".

5. In section 2, lines 86-87, you say "limiting the capability of robot to keeping only farmers on farm safe (either greenhouses or open-fields)". I don't understand what you mean in this sentence. Please, rewrite to make it clearer.

6. When the subject of a sentence is a paper, you must name the authors, not just the number of the reference. For example, instead of "[21] proposed...", it should be: "Lucet et al. [21] proposed...". Instead of "[22] present...", it should be "Meshram et al. [22] present...". Please review and correct all other similar cases.

7. Figure 1 appears before it is mentioned in the text. All the figures and tables should be placed after they are mentioned in the text, not before.

8. For all the commercial equipment referred in lines 199-200, you must indicate: model (manufacturer, city, country).

9. Please, rewrite the sentence in lines 206-207: "The frame of images from each views of the calibrated RGB stereo camera is proposed to provide the sensing for potential agrobot." This is incorrectly expressed. I don't know what you mean. Please, rewrite this sentence.

10. In Figure 2, instead of the labels "Human Vision Mechanism" and "Proposed Robot Vision Mechanism", you should put labels (a) and (b), and describe them in the caption. Moreover, I don't understand why the farsighted section of the image is the upper part, and the nearsighted is the lower part. What is the criterium to limit the upper and the lower part?

11. In subsection 3.3 you say that: "This relative insect pest scouting technique of the robot is motivated by the flocking of Bubulcus ibis around cattle and farm machinery to feed on insects during preplanting operations". I don't know what the relationship is between your technique and "the flocking of Bubulcus ibis around cattle". Please, explain better how this can motivate your work.

12. In section 3, the camera system used should be explained in more detail. You indicate that you use "off-the-shelf" cameras, but you don't mention the camera model used. So, you should describe the camera model used, the camera optics, the resolution of the cameras, the distance between the two cameras, etc., and if possible, include a picture of the camera system.

13. Why are there so many blank lines between lines 255 and 258? Please, remove them.

14. In line 293, you say that "the most accurate base network" is VGG19. However, this is a result of the experimental section. At this point, you should only say that you have tested 8 proposed classifiers and selected the most accurate one.

15. The idea behind the use of far and near-sighted sections should be explained and justified better. Why the far-sighted is the upper part? This depends on the location of the camera and movement of the robot. For example, in Figure 5, the images in the lower row show top views of the soil; so, the distinction between far and near-sighted is not correct. Moreover, what is the vertical limit to separate far from near? How did you fix it? Also, what happens if there are insects in the far section but not in the near section? What happens if there are insects in the near section but not in the far section?

16. In Table 1, the list of methods should not be present as a table in this format. I suggest you replace it with a table with the columns: Method, Reference, Description. Also, in the caption, instead of "List of insect classifiers", you should say that this is the list of standard methods used in the experiments for insect classification.

17. In line 316, you say that "Table 1 presents the feature extractors selected for the experiments". However, Table 1 is the list of insect classifiers. Do you mean that the same classifiers were used as feature extractors? This should be indicated clearer in the text.

18. The dataset of images and videos should be described better. You only say that "we only collected 862 images and some videos". But, how many videos? How many of the images contain worms? How many worms have been labelled in total? What is the resolution of the images in pixels? Also, since you have used a stereo system, do these 862 images correspond to 431 scenes (that is, 431 for the left camera and 431 for the right camera)? Or are they only 862 images from a single camera? Do you have the ground-truth 3D position of the worms in the images? Since the testing is performed on the videos, they should be described better: total number of videos, duration, resolution, total number of labelled worms, etc.

19. Regarding the dataset shown in Figure 7, it is very appreciated to give public access to your dataset. However, this dataset of images does not meet the conditions of the practical application of the system. It is supposed that your system is going to be applied on a robot looking to the soil. But many of the images are not in those conditions. For example, in the videos the worms are always in the soil, but in the dataset many of them are on leaves. Some images are cropped form-fitting (e.g. master/armyWorm/252.jpg), which is not adequate for training. On the other hand, you should also provide public access to the videos used in the experiments.

20. Regarding section 4, what is the GPU used in the experiments?

21. In Figure 8 and 9, you should use the same color and shapes for the same methods, in order to avoid confusion. Also, you should place tables and figures in the order that they are mentioned in the text, i.e. first Table 2, and then Figure 8 and 9.

22. In Figure 10 and 11, the labels ((a), (b), (c), etc.) should be centered. Also, the caption of Figure 11 should be extended to explain more clearly what (a), (b), (c), etc. mean.

23. In subsection 4.2, what is CoppeliaSim? Please, provide references.

24. In line 394, why "ELP 1.3 megapixels ELP-960P2CAM-LC1100 OV9715" is in blue color?Is this an off-the-shelf camera?

25. In the cosimulation (4.2), how are the images of the cameras obtained? This is not explained at all in the manuscript. I supposed they are synthetic images, but how are they generated?

26. In the conclusions, you say that the proposed method is "inspired by a natural enemy called Cattle-Egret". It is surprising that you have only mentioned this Cattle-Egret in the conclusions, and not in the description of the method. You should explain how this Cattle-Egret inspires your method.

27. Honestly, I think the proposed approach for worm location and neutralization is of no practical use. According to public data, Nigeria has an arable land area of 34 million hectares. Imagine a small robot that takes 65 seconds to analyze a small area of maybe 1 m2, which stops when it founds a worm to kill it. We would need about 1 million years!! Or we would need 1 million robots for a whole year!! This aspect should be discussed in the conclusion. It is true that the method could be used to detect and locate worms, but a more effective strategy would be required to eliminate the worms.

28. After the conclusions, the "Data Availability Statement" and "Conflicts of Interest" statements are missing.

29. The syntax, grammar, and spelling should be fully checked, as there are many errors in the manuscript. For example, I have noted the following:

L40. Thus, cannot be used -> Thus, they cannot be used

L43. the use of off-the-shelf RGB stereo camera -> You can say "the use of an off-the-shelf RGB stereo camera" or "the use of off-the-shelf RGB stereo cameras", but not "the use of off-the-shelf RGB stereo camera".

L56 and L225. infront -> in front

L57. of robot -> of a robot

L60. a accurate -> an accurate

L66. does not requires -> does not require

L93. Agricultural robot platform -> An agricultural robot platform

L95. labouratory -> laboratory

L96. support vector machine -> support vector machines

L102. present -> presents

L140. project, aimed -> project aimed (remove the comma)

L158. we propose the formal using -> we propose using

L162. The strategy take advantage -> The strategy takes advantage

L191. that provide -> that provides

L191. object in image -> objects in images

L203. is -> are

L224. to follows -> to follow

L276. a few datasets -> a small dataset

L291. l ies -> lies

L326. nonmaxima -> non-maxima

L387. on agricultural robot -> on an agricultural robot / on agricultural robots

L388. this usability -> its usability

L402. provide -> provides

Reviewer 2 Report

I find this a good work but if possible I would move the "related works" section adding it to the "results" section as discussion with proper modifications. In this way the section should be named "results and discussion".

Otherwise you may simply add a discussion in the "results" section to improve it.

I would add in the "conclusions" section some possible concrete applications for the farmers and possible future perspectives.

Good luck for you manuscript.

Reviewer 3 Report

Insects and plant pathogens cause substantial agricultural crop loss each year.  Robots, AI, sensors and other digital agriculture technologies can make farms more resilient by identifying specific areas of a crop field that require chemical treatments.  However, extant technologies detect infestations too late, often after a crop has been lost for the growing season.  This paper presents a digital agriculture solution, i.e., deep learning neural network and in-field robot, that detect Fall Armyworm in its larvae stage.  By detecting this pest earlier, more treatment and mitigation options are available.  However, Fall Armyworm are microscopic in larvae stage and largely camouflaged with the crop.  This paper employs off-the-shelf stereo camera systems to capture crop images in 3D with high resolution.  The paper contends that the additional dimension (compared to 2D images) improves accuracy.  To evaluate the proposed approach, a dataset of 862 images was collected and labeled (with expert guidance).  The data is available online.  The approach was evaluated using a variety of deep learning architectures, showing the VGG-16 and VGG-19 architectures were well suited for the task.

This was an excellent paper.  The paper could probably use a round of editing for posterity, but the ideas are clear and the long-term contributions are valuable.  The following comments are intended to inform future work.  First, I am concerned by the "Accuracy" reported for the classifiers.  95+% accuracy while achieving 30% recall suggests an imbalanced dataset.  The authors can read "Assessing the efficacy of machine learning techniques to characterize soybean defoliation from unmanned aerial vehicles" by Zhang et al to see approaches to deal with imbalance.  This matters because, in practice, the decision "not to spray" is consequential.  It is important to achieve high recall so that the pest doesn't reestablish after treatment.  Also, the reviewer would have appreciated an empirical comparison of Stereo RGB to 2-D RGB.  While the images make a clear argument for stereo, the reviewer wanders about the exact degradation caused by 2D RGB instead.

In conclusion, this is a very good paper.  Thank you for making the dataset open and available via Github.

Round 2

Reviewer 1 Report

The authors have done a great work answering all the questions and making the necessary changes to the manuscript. Consequently, the paper has significantly improved, and I consider it is acceptable for publication.